# Geological Approach for Landfill Site Selection: A Case Study of Vršac Municipality, Serbia

Ivana Carević, Mikica Sibinović 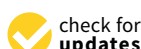, Sanja Manojlović, Natalija Batoćanin *, Aleksandar S. Petrović 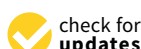 and Tanja Srejić

Faculty of Geography, University of Belgrade, Studentski trg 3/3, 11000 Belgrade, Serbia;
ivana.carevic@gef.bg.ac.rs (I.C.); mikica.sibinovic@gef.bg.ac.rs (M.S.); sanja.manojlovic@gef.bg.ac.rs (S.M.);
aleksandar.petrovic@gef.bg.ac.rs (A.S.P.); tanja.srejic@gef.bg.ac.rs (T.S.)
* Correspondence: natalija.batocanin@gef.bg.ac.rs

**Abstract:** One of the biggest problems of environmental protection in Serbia is landfills. It is often a case that the economic interests are predominant in the landfill sitting; thus, most landfills are not located according to standards. This study shows that detailed geological data assets combined with geographical modeling represents a reliable way to define and locate the landfill site. Geological evaluation is discussed in detail with regard to bedrock lithology, quaternary geology, geological structure, hydrogeology, surface runoff patterns, and topography. An approach combining geographical modeling and geology is presented for determining the sites suitable for landfill selection with respect to their geologic favorability. As opposed to numerous research papers on this topic, in the methodological procedure, special importance is devoted to the analysis of the geological criteria. In this way, it is significantly easier to determine the landfill area with the best characteristics due to geological structure and lithology which unequivocally and precisely indicates inadequate territories for candidate sites. The multicriteria decision analysis (MCDA) is based on geological criteria upgraded with road (primary, residential, secondary, and tertiary), settlements network, railway, airport, infrastructure, land use, hypsometry aquifer, wetland, and surface water. The score values are divided into four classes, i.e., restricted areas, suitable but avoid, suitable, and most suitable. Combining geographical modeling with geology led to the recognition of two locations to be most favorable for landfill site located in the most suitable area, which represents 25.3% of the study area.

**Keywords:** MCDA; geology; geological criteria; geographical modeling; land

## 1. Introduction

The environmental impact of landfill sites has led in the last few decades to increasing awareness of the geological factors in site selection, especially the potential for pollution [1]. Sanitary landfills are the most environmentally appropriate final destination of waste, but it is recommended that only waste with no potential for reuse, recycling, and recovery of energy should be disposed of at such sites [2,3] Most of the landfill sites across the world are old and are not engineered to prevent contamination of the underlying soil and ground water by the toxic leachate, which is defined as a liquid effluent containing contamination materials percolating through deposited waste and released within a landfill [4]. Thus, the worst condition for landfilling is to dispose the waste on the soil without any protection, lacking used compaction activities and engineered systems to biogas and leachate capture [5]. An understanding of surficial geology is therefore necessary and represents the starting point in initial selection of the potential landfill site [2–5]. The site's ability to isolate solid and liquid waste is also entirely determined by its geologic conditions [6]. The main geological criteria to be considered in landfill-site selection are the depth of soil and weathered rock, the occurrence of ground water and the potential for ground water or surface water pollution by leachate developed in the landfill [1]. Factors that must

be considered in evaluating potential sites for the long-term disposal of waste include: (1) haul distance, (2) location restrictions, (3) available land area, (4) site access, (5) soil condition and topography, (6) climatological conditions, (7) surface water hydrology, (8) geological and hydrogeological conditions, (9) local environmental conditions, and (10) potential ultimate uses for the completed sites [7,8]. These factors are also included in the law on waste management in the Republic of Serbia and are in accordance with criteria specified by relevant criteria for determination of the location of landfill site published in the Official Gazette of the Republic of Serbia No. 54/1992 and the regulation on waste disposal published in the Official Gazette of the Republic of Serbia No. 92/2010.

In recent years, a large number of studies has been published on landfill site selection using geographical information system (GIS) with multicriteria decision analysis (MCDA) with many different and often conflicting criteria applied [9]. Thus, landfill site selection may become more complex and difficult [10]. The application of geological criteria to waste disposal problems is significant and very convenient with respect to site selection solutions. Because of that, in this research, we combine geographical modeling and geological evaluation on more detailed criteria with stratigraphic, structural, lithological, magmatogenic, metamorphogenic, and geomorphological consideration. Igneous and metamorphic rocks are often in studies concerning landfill site selection considered as impermeable units because researchers do not take into account the structural and lithological criteria. In this regard, the reliable estimation of fractured igneous and metamorphic rocks is often lacking. Igneous and metamorphic rocks have very little primary porosity but may have considerable secondary porosity in joints and fractures. As a result, leachate flow may be fast. In this study, we use numerous geological criteria to accommodate requirements to evaluate the satisfactory landfill site.

The selection of waste landfill is one of the most important steps in managing urban solid waste [11]. The process of determining a suitable landfill area is extremely complicated because the site selection depends on many different factors and has an enormous impact on the economy, ecology and the environmental health. Economic factors include the costs associated with acquisition, development, and operation of the landfill area, which are closely connected with the agricultural land use. Demographic factors such as population densities, public health concerns, and settlement network systems are also hard to overcome. Environmental factors must be carefully treated to avoid contamination. Nonscientific and inappropriate disposal practices have a negative impact on the environment which affects the quality of life [12]. Therefore, in accordance with the reality that many factors must be incorporated into landfill siting decisions, GIS is very reliable for the preliminary researching because of the ability to manage large volumes of spatial data from a variety of sources, e.g., [13–29].

Several studies concerning selection of new sanitary landfills have been completed in recent years in the Vršac municipality (Figure 1) [30–34], which have not resulted in an optimal location for landfill site selection. The currently operating landfill in the study area is located NE from the city of Vršac in Mali Rit and covers an area of 26 ha. It is situated near the Vršac airport and Nature Park Mali Vršački Rit of the second category. This is not in accordance with landfill site selection near the area of outstanding natural beauty. In addition, landfill leachate has the potential to pollute the ground water recharge at the Vršac Mts. to the east (Figure 2). According to the National Waste Management Strategy, the city landfill does not meet the minimum protection measures and should be immediately sanitized, closed, and recultivated [33]. In 2014, a sorting and recycling center was built next to the city landfill, and its surface is two thousand square meters (Figure 3). During the past years, the Municipality of Vršac has undertaken a series of measures in order to rehabilitate the city landfill. This primarily refers to waste compaction by bulldozing and covering by earthen material. In the landfill area, part of the sanitized area is fenced and equipped with a minimization facility—pressing and baling waste that has the properties of secondary raw materials. However, the measures taken are not sufficient to prevent

the negative impact of decades-long waste disposal, especially when it comes to pollution caused by the leachate e.g., [30,31].

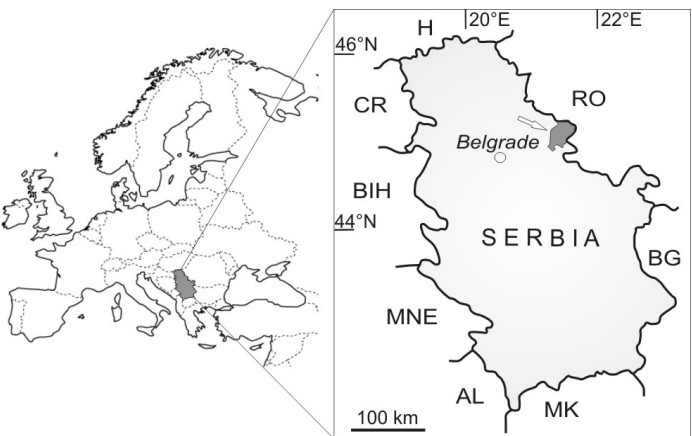

**Figure 1.** Location map of the study area (white arrow indicates the position of Vršac municipality).

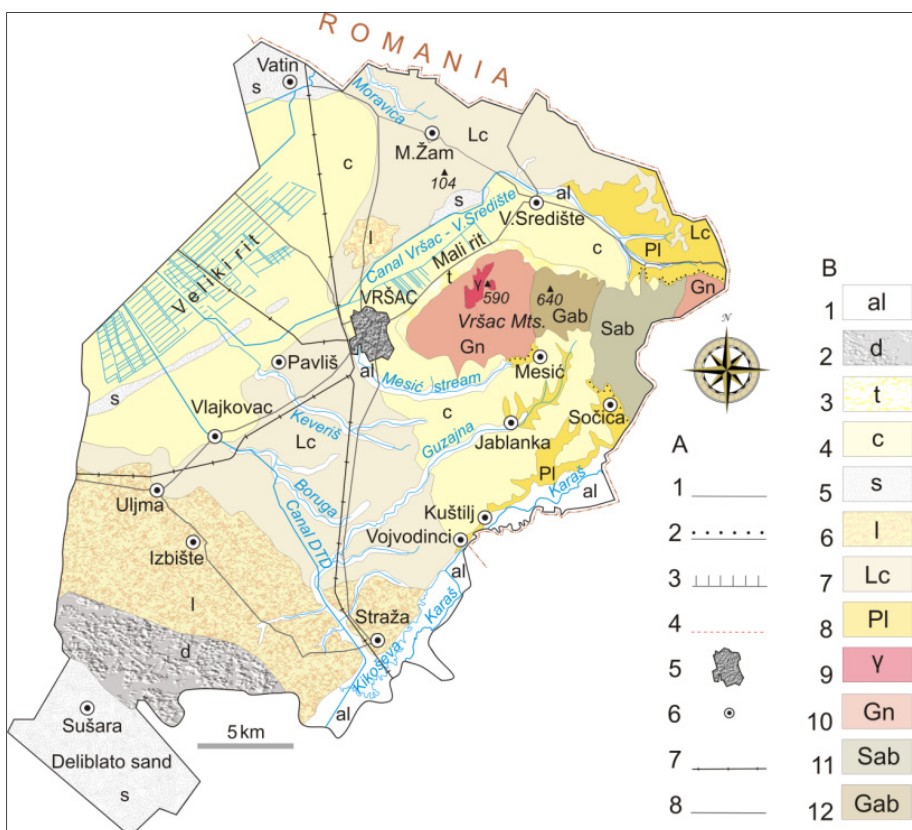

**Figure 2.** Simplified geological map of the Vršac municipality based on the Basic Geological Maps of Former Yugoslavia: sheet Vršac, 1:100,000 [34] and sheet Bela Crkva, 1:100,000 [35]. A: 1—Established geological boundary; 2—Established erosion or tectonic-erosion boundary; 3—Established boundary of pluton intruded in the surrounding rocks; 4—Assumed fault; 5—City; 6—Village; 7—Railway; 8—Road. B: Holocene: 1—Alluvial deposits; 2—Deluvium; 3—Talus; 4—Clay; 5—Sand; Pleistocene: 6—Loess; 7—Loess clay; Pliocene: 8—sand, sandstone, conglomerate, clay and marl; Late Paleozoic: 9—Granite; Precambrian: 10—Gneiss; 11—Albite-muscovite schist; 12—Albite gneiss.

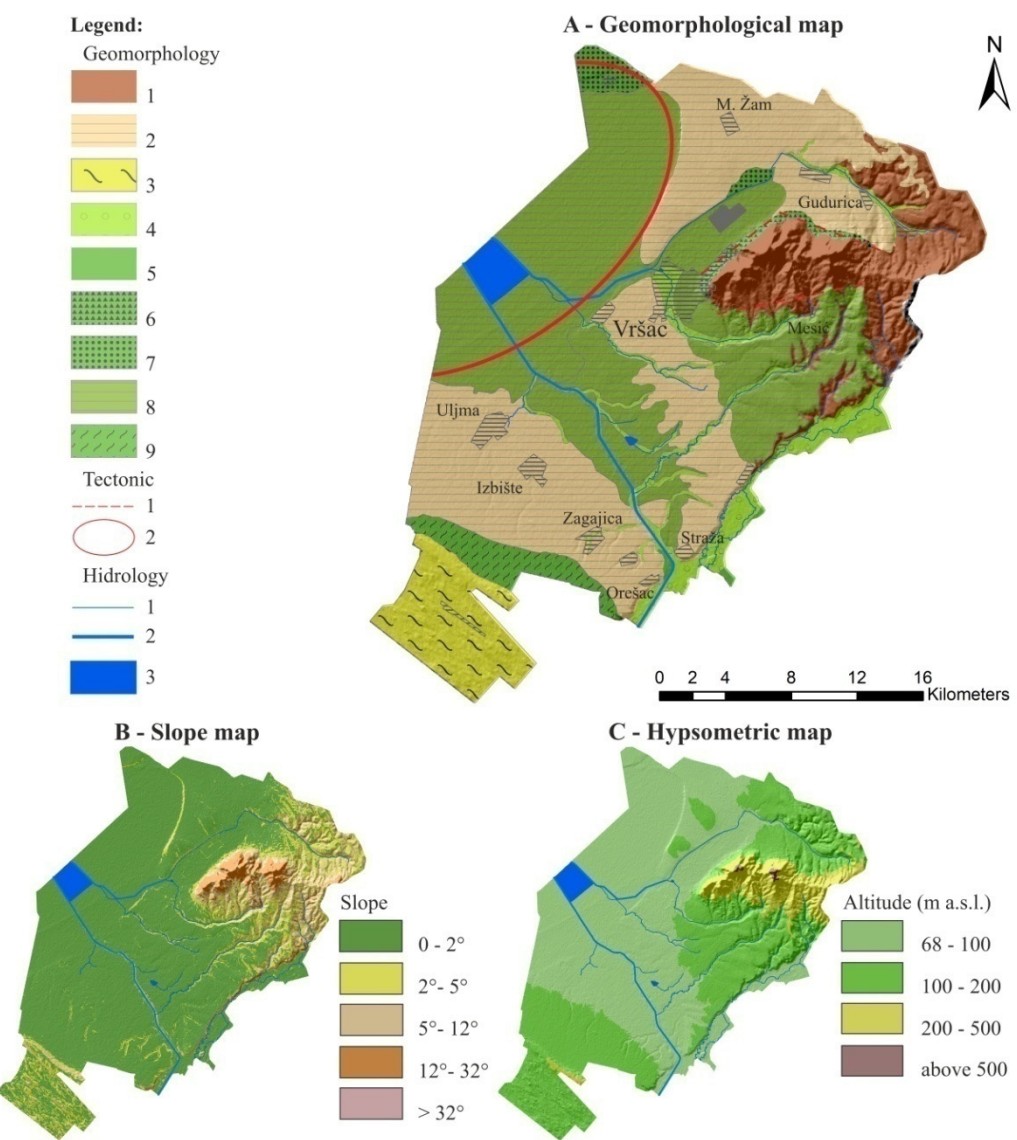

**Figure 3.** Simplified geomorphological map of the Vršac municipality: (**A**) Geomorphology: 1—Slope erosion area; 2—Loess plateau; 3—Banatska sands with parabolic dunes; 4—Alluvial plane; 5—River terrace; 6—Talus cones; 7—River sand depositions; 8—Upper Pleistocene river terrace; 9—Deluvial fans. Tectonic: 1—Fault; 2—Tectonic depression; Hidrology: 1—River; 2—Channel; 3—Fishpond. (**B**) Slope map and (**C**) hypsometric map.

This study aims to define the requirements of landfill site selection related to various aspects of geological criteria and environmental requirements on the basis of research that has been conducted in Vršac municipality. The geological approach for landfill site selection used in this study provides a more complex point of view than previous studies in this area due to the fact that geological criteria primarily control the suitability of landfill sites. The importance of bedrock is stressed as contaminated leachate may readily percolate downwards into the ground water depending on geological setting. A new multidisciplinary approach is therefore needed for landfill site selection as geological criteria primarily control the suitability of candidate sites. It is important to highlight the fact that geological surveys need to be supported by the ecological and social considerations and reasonable site management. Thus, the best results in landfill planning may be achieved by the collaboration of spatial planners and geologists in order to ensure that all factors affecting the suitability of sites for landfill site are taken into consideration. The relation between GIS analysis and MCDA based on geological standards upgraded with other

environmental and anthropogenic criteria outlined in the paper can provide a good and reliable model for successful landfill site selection and serve as a model of landfill siting for the territory of Serbia.

*Literature Review*

Numerous of authors have used multicriteria decision analysis (MCDA) in combination with other methods to solve waste disposal problems. The results of their research indicated that the use of MCDA, GIS, and remote sensing to identify landfill site selection is much more efficient from conventional methods [36,37]. Consequently, many methods for landfill sites integrate MCDA with GIS [38–40]; therefore, Şener et al. [18] adopted an integrated MCDA based on GIS to provide an effective tool for solving landfill site selection problems. On the one hand, GIS enabled better data manipulation and presentation, while on the other hand, the MCDA consistently ranked potential landfill site selection based on different criteria. In addition, the need for the use of MCDA in solid waste management systems was discussed by Cheng et al. [40], as these frameworks could have complex and inconsistent impacts on various stakeholders [12]. The benefits of MCDA local authors have also been stated in their research: Popović et al. [30] accentuated the benefits of MCDA for regional landfill and recycling center site selection; Kostić et al. [41] point up the importance of multicriteria decision analysis for local management plan for communal waste; Vidović and Gordanić [42] underlined needs for integration of MCDA with hydrogeochemical investigations; Lukić et al. [43] emphasized geomorphic diversity, while Vušković et al. [44] used multicriteria decision analysis for better understanding of wastewater treatment concept. In accordance with an extensive literature and a wide range of research areas, a GIS-based MCDA can be used to define landfill site selection and precise mapping of potential sites.

## 2. Study Area

The terrain which covers an area of Vršac municipality (Figure 1) is situated in the southeastern Banat on the border with Romanian counties Caraş-Severin and Timiş. It covers an area of about 800 km$^2$ and represents the extreme southeastern part of the Pannonian Basin. The south corner of the municipality is Deliblato Sands, whereas the most striking part of the relief occupies Vršac Mts. with Kudrič Peak (640 m). The relief of the terrain is typically flat, with absolute altitudes of 75–100 m a.s.l. The northern and eastern border of the municipality is the state border with Romania. The Kikoševa and Karaš rivers flow in the eastern area. The Boruga, Keveriš, and Mesić streams, as well as the Danube-Tisa-Danube canal drain the central part, while the Moravica River and Vršac-Veliko Središte canal flow through the northern area of the Vršac municipality (Figure 2). In the municipality prevails a mild continental climate. Winter is temperately cold, and summer is dry and warm.

The area of the Pannonian Basin in Serbia is characterized by weak seismicity with an irregular distribution of epicenters, but on the other hand, the southern margin of the Pannonian Basin is the most active area of Serbia in the seismic sense [35]. Based on previous seismic activity and maps of microseismic reonization of Vojvodina Province, the area is endangered by the earthquake of 7 MCS for a return period of 100 years [33]. This is in agreement with statement given by Toth (2008) for the Pannonian basin [41].

### 2.1. Occurrence of Ground Water

Ground water occurs in Pliocene and Quaternary alluvial aquifers situated near the high-grade metamorphic complex border of Vršac Mountains. Ground water recharge is mainly by the infiltration of rainfall into fractured rock in hilly areas.

Ground water is assigned on the basis of chemical analyses to the group of hydrocarbonate-calcium-magnesium noncarbonated low-mineral water, whose physical properties and chemical composition is the consequence of geological conditions of the

environment through which circulates. There are two main ground water resources in the area: the Mesić and Pavliš water sources.

The dominant anions and cations ($HCO_3^-$, $Ca_2^+$ and $Mg_2^+$) accumulate in ground water of Mesić as a product of weathering of silicate minerals present in igneous and metamorphic complex of Vršac Mountains [45]. Water was found at a depth of 50–65 m. Deep ground water table (at least 50 m) in region is suitable to avoid pollution from the surface [4,46].

The main resource of ground water for drinking-water supply of the population and industry are the aquifer, which was formed in Quaternary river—lake gravel—sandy deposits in the water source Pavliš. The captured aquifer layers of sand and sandy gravel are found at depths of over 30 m and have a thickness of up to 60 m. The average thickness of the aquifer complex in the area of the water source Pavliš is about 52 m. Average exploitation is estimated at 150 L/s, and the quality of water meets the standards for drinking water. Recharge of aquifer is performed by direct infiltration of rainfall in water-bearing deposits around the slopes of the Vršac Mountains and the wider area of the town Bela Crkva [46]. On territory of the Vršac municipality, there is also the Straža water source that supplies the settlements in the municipality Bela Crkva [47].

Centralized wastewater treatment of the settlement is a dominant world practice. Such a system of treatment process possesses only a few cities in Vojvodina (Vršac, Sombor and Subotica). The total quantity of wastewater from households and industry in Vršac is 283 $m^3$/ha. For now, these waters, after treatment processes, are disposed of into the Danube-Tisa-Danube canal [48].

*2.2. Geological Setting*

According to stratigraphic range, the oldest rocks are represented by Precambrian gneiss and albite-muscovite schist cropping out in the Vršac Mts. (Figure 2). These correspond to the low-grade metamorphic complex of Văradia Mts. in Romania [49,50]. The metamorphic series thickness is inferred as about 6000 m. During the late Paleozoic, the granitoid massif was intruded [49]. The rocks of Mesozoic age were not recognized in the study area.

Relatively thick succession of genetically different types of Pliocene and Quaternary sediments transgressively overlies metamorphic complex. Pliocene sand, sandstone, conglomerate, clay, and marl are recorded in eastern corner and reach the thickness of about 2000 m.

Pleistocene and Holocene deposits cover more than 90% of the municipality surface. These Quarternary formations are of fluvial and aeolian origin and were deposited at the time when Pannonian basin became mainland, with river, lakes, and puddles [51]. Pleistocene loess and loess clay occur to the north and central-south area. The loess attains a thickness of 25 m in the south, but it is only 5 m thick in the north. Loess as semiconsolidated sediment has originated by the accumulation of aeolian dust predominately over the initial fluvial bottom of the Pannonian Basin. Paleosol is recorded in boreholes at an average depth of about 15 m. The paleosol thickness is inferred as 0.5–2.4 m [52]. Loess clay overlies the loess and reaches the thickness about 50 m. Holocene sand are observed in northwestern and southern corners of the municipality. The thickness of these sediments reaches about 10 m. Clay occurs on the west in Veliki Rit, in central part in Mali Rit as also to the east. It reaches a thickness of only 5–10 m dominated by clayey and sandy particles. Clays have been deposited in ponds and swamps in Vršac depression. Diluvium occur in southern hilly terrain. Talus deposits overlie weathered Precambrian gneiss bedrock. Sand and gravel alluvial deposits are well exposed in the eastern area, whereas in the major river, courses attain the largest extent. In many examples, these materials constitute essential high-yield aquifers.

Faults are important features to note as a point of view when selecting the landfill site [42]. Active faults were not observed in the territory of municipality. Assumed faults are presented only in the granitoid pluton in fractured low-crystalline schist of Vršac Mts

and along the Karaš River course. This fractured metamorphic complex influences ground water recharge. However, a geophysics survey in the northern area of municipality has shown a set of faults at a depth of 300 m. This system played a significant role in the subsidence and formation of Banat depression [53].

*2.3. Geomorphological Setting*

Vršac municipality has two clearly height-differentiated units in relief: a mountainous and a Pannonian part (Figure 3C). The higher unit consists of the Vršac Mountains, one of the two independent mountain massifs in Vojvodina. On them is the highest peak of Vojvodina, Gudurički vrh (641 m a.s.l.). A wide mountain ridge of Vršac Mts. is divided by four peaks. Due to the significant slope of the terrain (Figure 3B), the mountainous sides are dominated by the slope process (Figure 3A). In addition to the slope, the favorable lithology (loess and clay) in the north and south footslope enabled the development of strong gully erosion [54]. The footslope on the western mountain side is characterized by deluvial sediments in the form of talus cones.

The lower, Pannonian part of the research area consists of several geomorphological units. The largest area is occupied by the South Banat loess plateau and the Upper Pleistocene river terrace. The South Banat loess plateau is presented in two parts. In the southern part, there is an easy plateau of Dumača with pseudokarst sinkholes in loess up to 300 m in diameter [43]. This loess plateau ends with an escarpment 15 m high toward the Alibunar depression. Around the Vršac Mountains, from Atskagreda in the north to Straža in the south, there is a plateau built of loess and loessoid loam.

Between the loess plateaus, there is a Vršac–Alibunar tectonic depression (Figure 3A). In this depression and the river valleys that are open to it, the Upper Pleistocene river terrace was built (Figure 3A), by mixing marsh sediments and loess blown from the surrounding plateaus [55].

The southwestern part of Vršac municipality belongs to Banatska peščara Sands (Figure 3A), a part called "high sands" [56,57]. The Banatska peščara Sands is characterized by a distinctly dune relief, which represents the long dunes of the NW-SE direction and the interdepression [47]. The height difference between dunes and depressions is 20–30 m. The sand accumulated in the Banatska peščara sands comes from the alluvial material made by the Danube floods during the Holocene [55]. This sand is mostly driven by prevailing winds that blow from the SE quadrant (Košava wind) and form more than 1300 parabolic dunes [58].

The youngest forms in the relief are alluvial plains, formed along the rivers flowing from the Vršac Mountains (Figure 3A).

## 3. Materials and Methods

In this study, two candidate sites for an appropriate landfill area are analyzed and compared with the current position of official landfill area by using geographic information system (GIS) and multicriteria decision analysis (MCDA). Compatibility of GIS and MCDA to solve the landfill site selection problem has been often used in researching. GIS provides efficient presentation of the data, and MCDA supplies consistent ranking of the potential landfill areas based on a variety of criteria [59–67]. A multicriteria decision analysis (MCDA) and weighted overlay analysis [18] were used for the suitable landfill site using GIS environment [61]. The preferences of decision makers depend on the relative importance of options according to a number of criteria defined by experts [59–67]; therefore, in this study, MCDA sets a preference ordered class for 14 variables. In the decision-making process, the integration of spatial data was treated as an important analytical method for defining and solving the problem of multiple decision making. In accordance with this technique, various thematic layers have been generated and integrated to develop the best landfill sites. In relation to a particular attribute of interest, MCDA sets theory uses a membership function that characterizes the degree of membership value; therefore, an interesting attribute is calculated at discrete intervals, and the membership function can be

presented as a table to classify the map according to membership values. Thus, reducing the negative impact on the environment, ecology, and economy is the basic meaning of deciding on the best location for landfills.

Integrating MCDA and GIS is a great contribution that usually yields very useful spatial alternatives to help decision makers [68]. In the past, analytic hierarchy process (AHP) was one of the useful methodologies in landfill selecting, because it allows group decision making, where group members can use their experience, values, and knowledge to break down a problem into a hierarchy [69,70]. The role of the AHP is to deconstruct a problem, in a hierarchical context, at a level where the data are compared in pairs in order to assess the weight of each, at the next level [71]. On the other hand, in recent studies, many authors try to integrate GIS and fuzzy multicriteria decision analysis (FMCDA) based on a spatial decision support system (SDSS) for waste management [72–78]. The basic problem that appears in most of studies is the neglect or inadequate research of the geological composition as a crucial factor in the planning of the landfill site. In this study, multicriteria decision analysis (MCDA) is based on geological criteria which are upgraded with road (primary, residential, secondary and tertiary), settlements network, railway, airport, infrastructure, land use, hypsometry aquifer, wetland, and surface water. In accordance with this method, the 14 map layers defined a criterion which has to be considered in landfill site selection. This criterion has a different scale and must be standardized to a common dimensionless unit in each map layer. In the score range procedure, the standardized scores are calculated by dividing the difference between the maximum raw score and a given raw score by the score range [18,79]. Simple additive weighting method is the most often used as a multiattribute decision technique based on the weighted average:

$$X_{ij}^1 = \frac{X_j - X_{ij}}{X_j{}^{max} - X_j{}^{min}},\tag{1}$$

where $X_{ij}^1$ is the standardized score for the $i$th alternative and $j$th attribute; $X_{ij}$ is the raw score; and $X_j{}^{max}$ and $X_j{}^{min}$ are the maximum and minimum score for the $j$th attribute, respectively. This procedure is applied to each input raster in GIS. Normalized deviation is a measure of variation that shows the algebraic deviation of one value of the characteristic from the arithmetic mean, expressed in standard deviations. This measure is suitable for comparing the variation of features from different numerical series, of which characteristics are expressed in different units of measure. After the standardization of scores in each map layer, the criterion weights are defined and presented in the Table 1.

**Table 1.** The criterion weights defined for simple additive weighting method (SAW) [18,79].

| Data Layer | Weight | Normalized Weights |
| --- | --- | --- |
| Geology | 10 | 0.12511 |
| Urban center | 10 | 0.12511 |
| Villages | 9 | 0.11260 |
| Surface water | 8 | 0.10009 |
| Wetlands | 8 | 0.10009 |
| Land use | 7 | 0.08758 |
| Hypsometry | 6 | 0.07507 |
| Electricity | 6 | 0.07507 |
| Primary roads | 5 | 0.06256 |
| Secondary roads | 4 | 0.05005 |
| Tertiary roads | 3 | 0.03753 |
| Airport | 2 | 0.02501 |
| Residential roads | 1 | 0.01251 |
| Railways | 1 | 0.01251 |

The initial selection of optimal municipal waste landfill site in this study is in accordance with general geological/hydrogeological and topographic criteria specified by

relevant criteria for determination the location of landfill site published in the Official Gazette of the Republic of Serbia No. 54/1992 and regulation on waste disposal published in the Official Gazette of the Republic of Serbia No. 92/2010. Economic distances from waste sources and road networks were also considered.

The geological criteria used in the selection of waste disposal sites follow the parameters proposed by [80]. The site selection process was conducted through the assessment of following geological criteria: bedrock lithology (rock type, grainsize characteristics, and texture); quaternary geology (character, thickness, and homogeneity of sediments); geological structure (attitude of bedding, folding, faulting, and jointing, including discontinuities on all scales); hydrogeology (ground water levels, distribution of aquifers ground water flow patterns, etc.); surface runoff patterns (size and discharge of streams running through the site controlled by the topography of the site) and topography (shelter from wind and visual impact).

In order to protect against the risk of pollution, the location of waste disposal sites evidently requires the consideration of numerous criteria [81]. In the most circumstances, the geological, hydrogeological, and topographical parameters primarily control the technical suitability of landfill sites [82]. Geological assessment is based on detailed study of geological maps. The materials used in the study consisted of two sheets of the Basic Geological Maps of Former Yugoslavia: sheet Vršac, 1:100,000 [34] and sheet Bela Crkva, 1:100,000 [35]. The geological formations were grouped into twelve units according to lithology and stratigraphic range (Figure 2).

## 4. Results and Discussion

Assessment based in the study has indicated that bedrock formations exposed in the Vršac Mountains are generally not suitable for sanitary landfills. The Vršac Mts. were declared in 1982 as a landscape of outstanding features. Furthermore, these represent the zone of ground water recharge with common drainage to aquifer layers. The eastern area of the municipality should be avoided in landfill planning due to thick, mostly unconsolidated Pliocene deposits, Holocene alluvial deposits, and several natural monuments. Natural monument Straža is situated on the right side riverbank of Karaš. It represents the protected oak forest. The groove in Kuštilj is also declared as Natural Memorial Monument. Thick interbedded loess cropping out on the south and northwest from Vršac city should be avoided for landfill site due to high permeability. The same refers to Quaternary unconsolidated deposits which have the high permeability such as alluvial deposits and Deliblato sands declared as Special Nature Reserve. Because of the extremely high permeability of alluvial deposits, leachate may move freely and invariably contaminates contained waters [83]. If there is no appropriate natural barrier, the contamination of ground water is almost certain [84].

Currently operating landfill site is located in highly clayey zone which present problems in excavation and determining direction of leachate movement. Moreover, it is situated next to the Nature Park Mali Vršački Rit, airport, and ground water recharge at the Vršac Mts (Figure 4).

One geological unit in the Vršac municipality is generally suitable for landfill. On the north crops loess clay yield little water, into which leachate movement would be very slow (Figure 2). The landfill should be located in slowly permeable sediments. These deposits allow a partial renovation of leachate through various processes [85]. Compacted loess clay contains silty and clayey particles, reaches a thickness of 50 m, and has a wide distribution in the territory of the municipality. The most favorable potential landfill locations are located north from Vršac city with no permanent watercourses. In addition, no active faults were observed in this area.

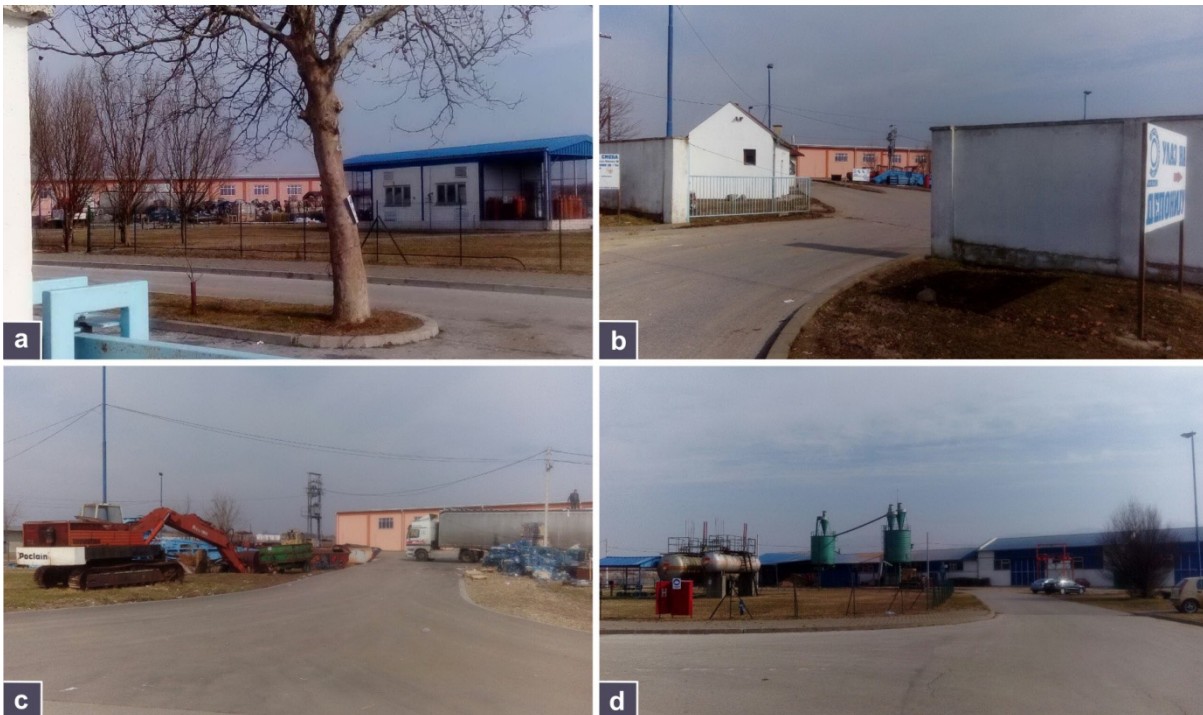

**Figure 4.** City landfill with sorting and recycling center. (**a**) Sorting center; (**b**) entrance to the landfill; (**c**) bulldozer used for waste compaction; and (**d**) recycling center.

In order to determine the suitable landfill site, the criterion weights are normalized to generate the overall score for each alternative then converted into map. The score values are divided into four classes, (1) restricted areas for landfill sitting, (2) suitable but avoid, (3) suitable, and (4) most suitable classes. The area belonging to most suitable class covers 25.3% of the municipality area and lies on the loess clay (Figure 2).

It is important to emphasize that the olandfill location is, according to this analysis, in the *suitable but avoid* area.

In accordance with these results, two candidate sites are determined for further detailed geotechnical and hydrogeological researching. *Candidate site 1* is compared with the Official landfill location 2.5 km western, at the same distance from the urban settlement (Vršac), at a safe distance from the surface water, on a favorable lithology and with satisfactory transport accessibility. *Candidate site 2* is located 5 km north of the official landfill location, at a safe distance from the surface water on a favorable lithology but without access road. This location provides the possibility of urban expansion of the city, which is not the case with *Candidate site 1* and official landfill location. The comparative advantages of alternative locations relative to the official landfill location are related to the geological criteria and the possibilities of spreading the settlements network system and are not at any potential risk caused by natural hazards.

## 5. Conclusions

Selection of the landfill site is a very important step in the construction of the landfill. In this study, all input data required for the analyses are generated from the map sources such as: geology, settlements network (urban center, villages), surface water, wetlands, land use, hypsometry, infrastructure, roads (primary, secondary, tertiary, and residential roads), airport, and railways. During the process of the landfill location selection, the political and financial criteria have not been considered. In addition to GIS modeling, a geological evaluation was undertaken in order to identify locations which have favorable geologic and hydrologic conditions for landfill site. The waste disposal is possible in two places of the municipality due to the GIS and MCDA analysis (C*1* and C*2*) (Figure 5).

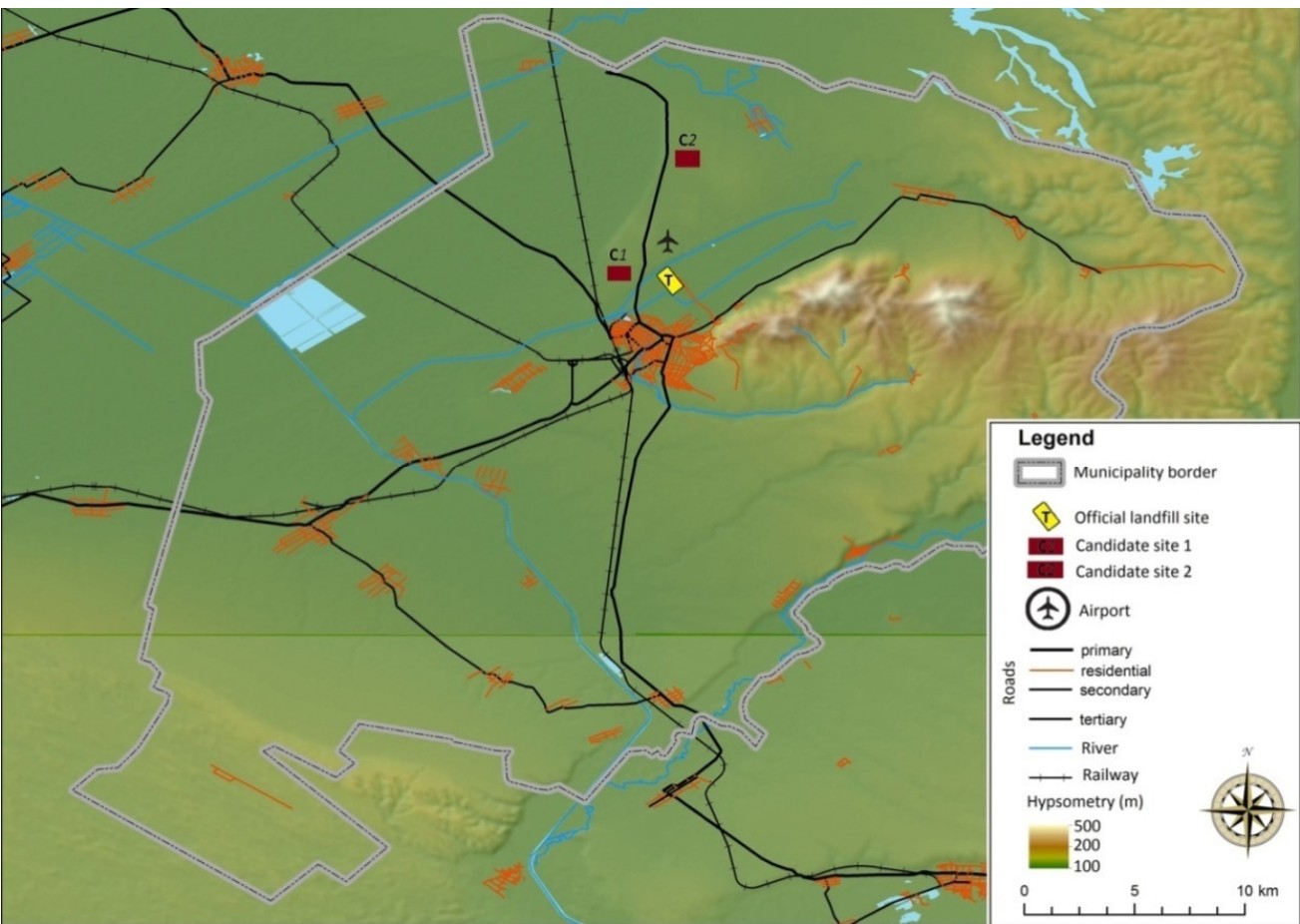

**Figure 5.** Geographical map of the Vršac municipality based on synthesis of the different layers.

Decision analysis was used in order to reduce potential risks of ground water contamination by landfill leachate. The selected landfill sites are not at any potential risk caused by natural hazards. There are no observed active faults; the relief of the terrain is typically flat and not affected by landslide; there are no permanent watercourses which may cause flood risk. Candidate sites satisfy requirements of the landfill sites and represent careful field checks which allow further geotechnical and hydrogeological researching and analyses for final site which would be able to replace today's official landfill location.

Sustainable waste management is a great challenge of modern man and requires an organized and coordinated set of different activities. In this context, adequate location of landfills becomes an integral part of the concept of sustainable development of an area. The results of this research show that only precise and scientifically justified determination of geographical and geological factors of the landfill location is a safe way to solve long-term environmental problems and achieve economic, social, and environmental goals of people in that area.

Finally, the Republic of Serbia's entry into the European Union depends on the many standards that it needs to meet. Moreover, ecology, regulated by the Chapter 27 in the EU accession, is one of the most important issues. The main problem in the area of landfill selection in Serbia so far is the failure to comply with regulations and the lack of legal sanctions. The results of this study might be of interest for future research in the field of environmental protection. Therefore, there is a necessary need to begin a nationwide geological survey to locate suitable landfill sites. In order to produce directories of such sites, a register of potential sites could be created to help spatial planners in selecting eventual future landfill sites.

**Author Contributions:** Conceptualization, I.C. and M.S.; methodology, I.C.; software, A.S.P.; validation, I.C., M.S., and S.M.; formal analysis, I.C.; investigation, M.S.; resources, N.B.; data curation, T.S.; writing—original draft preparation, I.C.; writing—review and editing, M.S.; visualization, A.S.P.; supervision, I.C.; project administration, T.S.; funding acquisition, N.B All authors have read and agreed to the published version of the manuscript.

**Funding:** This research was funded by Ministry of Education, Science, and Technological Development of the Republic of Serbia, project nioid200091.

**Institutional Review Board Statement:** Not applicable.

**Informed Consent Statement:** Not applicable.

**Data Availability Statement:** Not applicable.

**Acknowledgments:** This study is a research project funded by the Ministry of Education, Science, and Technological Development of the Republic of Serbia.

**Conflicts of Interest:** The authors declare no conflict of interest.

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
