# Peer review of "Geological Approach for Landfill Site Selection: A Case Study of Vršac Municipality, Serbia"

_sustainability, doi:10.3390/su13147810_

Round 1

Reviewer 1 Report

The authors have conducted a research for determining the most suitable place for landfilling. The research is interesting; my main concerns are the organization of the sections and subsections and the lack of justifications and elaborations in the literature review and methodology sections. My comments are as follows.

The abstract should be re-written. Please reorganize the sentences to enhance its readability. Start with the gap/ problem, followed by aim, methods, results, and implications.

The literature review is not sufficient and the criteria used for the research should be well justified either by experts or strong justifications based on an extensive literature review.

Sub-section 2.4 doesn’t seem relevant to be under the Materials and methods section. Moreover, sub-section 2.5 “research methodology” is not suitable to be under Section 2. I recommend authors reconsidering the structure of the sections and subsections.

The authors discussed the benefits of AHP, and FMCDA approaches, and then they have chosen simple additive weighting. Strong justifications are needed. Moreover, the steps taken to conduct the analysis should be clearly mentioned.

Where did you get the findings of Table 1? If it is a finding, why is it located in the research methodology section?

Section 3 begins with several citations. This section is “Results” so the authors should present the findings of the research.

Future research should be mentioned in the conclusion.

Author Response

POINT-BY-POINT RESPONSE TO REVIEWS

We are very grateful to the thorough revision by Reviewers, which helped us to clarify and, generally, to improve our early manuscript. We have considered carefully all the remarks and certainly appreciate the significant efforts to make our results more understandable to the readers. All the changes are highlighted in yellow in the revised manuscript to facilitate reviewer reassessment. Please, find below our responses to the reviews comments that are shown in red below each comment.

Response to Reviewer 1 Comments:

General comments: The authors have conducted a research for determining the most suitable place for landfilling. The research is interesting; my main concerns are the organization of the sections and subsections and the lack of justifications and elaborations in the literature review and methodology sections.

Response: Thank you very much for the excellent and professional revision of our manuscript. We have studied comments carefully and made correction which we hope meet with approval. We appreciate your invaluable and concrete suggestions, that we believe, have now greatly improved the manuscript. We provide our point-by-point responses as follows:

Point 1: The abstract should be re-written. Please reorganize the sentences to enhance its readability. Start with the gap/ problem, followed by aim, methods, results, and implications.

Response 1: Thank you. We have reorganized the sentences in abstract according to suggestion. It now reads as follows:

One of the biggest problems of environmental protection in Serbia is landfills. It is often a case that the economic interests are predominant in the landfill sitting, thus most landfills are not located according to standards. This study shows that detailed geological data assess combined with geographical modeling represents the reliable way to define and locate the landfill site. Geological evaluation is discussed in detail with regard to bedrock lithology, quaternary geology, geological structure, hydrogeology, surface runoff patterns and topography. An approach combining geographical modeling and geology is presented for determining the sites suitable for landfill selection with respect to their geologic favorability. As opposed to numerous research papers on this topic, in the methodological procedure, special importance is devoted to the analysis of the geological criteria. In this way, it is significantly easier to determine the landfill area with the best characteristics due to geological structure and lithology which unequivocally and precisely indicates inadequate territories for candidate sites. The multicriteria decision analysis (MCDA) is based on geological criteria upgraded with road (primary, residential, secondary and tertiary), settlements network, railway, airport, infrastructure, land use, hypsometry aquifer, wetland, and surface water. The score values are divided into four classes, i.e., restricted areas, suitable but avoid, suitable and most suitable. Combining geographical modeling with geology led to the recognition of two locations to be most favorable for landfill site located in the most suitable area which represents 25.3% of the study area.

Point 2: The literature review is not sufficient and the criteria used for the research should be well justified either by experts or strong justifications based on an extensive literature review.

Response 2: The Literature review is added as an Introduction subsection 1.1. It now reads as follows:

Numerous of authors have used multi-criteria decision analysis (MCDA) in combination with other methods to solve waste disposal problems. The results of their research indicated that the use of MCDA, GIS and remote sensing to identify landfill site selection is much more efficient from conventional methods [80,81]. Consequently, many methods for landfill sites integrate MCDA with GIS [82-84], so Şener et al. [85] adopted an integrated MCDA based on GIS to provide an effective tool for solving landfill site selection problems. On the one hand, GIS enabled better data manipulation and presentation, while on the other hand, the MCDA consistently ranked potential landfill site selection based on different criteria. Also, the need for the use of MCDA in solid waste management systems was discussed by Cheng et al. [86], as these frameworks could have complex and inconsistent impacts on various stakeholders [87]. The benefits of MCDA local authors have also stated in their research: Popović et al. [30], accentuated the benefits of MCDA for regional landfill and recycling center site selection, Kostić et al. [34] point up the importance of multi-criteria decision analysis for Local management plan for communal waste, Vidović & Gordanić [34] underlined needs for integration of MCDA with Hydrogeochemical investigations, Lukić et al. [44] emphasis geomorphic diversity, while Vušković et al. [53] used multi-criteria decision analysis for better understanding of wastewater treatment concept. In accordance with an extensive literature and a wide range of research areas, a GIS-based MCDA can be used to define landfill site selection and precise mapping of potential sites.

Point 3: Sub-section 2.4 doesn’t seem relevant to be under the Materials and methods section. Moreover, sub-section 2.5 “research methodology” is not suitable to be under Section 2.

 I recommend authors reconsidering the structure of the sections and subsections.

Response 3: Thank you. We reorganized the structure of Section 2. Materials and methods are now under the separate Section 3.

Point 4: The authors discussed the benefits of AHP, and FMCDA approaches, and then they have chosen simple additive weighting. Strong justifications are needed. Moreover, the steps taken to conduct the analysis should be clearly mentioned.

Response 4: The justification, analytic steps and decision-making process with MCDA sets are now more clearly mentioned and extended. It now reads as follows:

A multi-criteria decision analysis (MCDA) and weighted overlay analysis [18,74] were used for the suitable landfill site using GIS environment [56]. The preferences of decision-makers depend on the relative importance of options according to a number of criteria defined by experts [54-62], so in this study MCDA sets a preference ordered class for 14 variables. In the decision-making process, the integration of spatial data was treated as an important analytical method for defining and solving the problem of multiple decision-making. In accordance with this technique, various thematic layers have been generated and integrated to develop the best landfill sites. In relation to a particular attribute of interest, MCDA sets theory uses a membership function that characterizes the degree of membership value, so an interesting attribute is calculated at discrete intervals and the membership function can be presented as a table to classify the map according to membership values. Thus, reducing the negative impact on the environment, ecology and economy is the basic meaning of deciding on the best location for landfills.

Point 5: Where did you get the findings of Table 1? If it is a finding, why is it located in the research methodology section?

Response 5: Table 1. (The criterion weights defined for simple additive weighting method) is a part of methodology explanation, so we add the relevant references to which this method refers.

It now reads as follows:

Table 1. The criterion weights defined for simple additive weighting method (SAW) [18, 74].

Point 6: Section 3 begins with several citations. This section is “Results” so the authors should present the findings of the research.

Response 6: We removed the citations to the last paragraph of the former Section 2.5 and start the Results and Discussion chapter as:

Assessment based in the study has indicated...

Point 7: Future research should be mentioned in the conclusion.

Response 7: We added more information in the Conclusion on the future research as follows:

Finally, the Republic of Serbia’s entry into the European Union depends on many standards that it needs to meet. And ecology, regulated by the Chapter 27 in the EU accession, is one of the most important issues. The main problem in the area of landfill selection in Serbia so far, is the failure to comply with regulations and the lack of legal sanctions. The results of this study might be of interest for future research in the field of environmental protection. Therefore, there is a necessary need to begin a nationwide geological survey to locate suitable landfill sites. In order to produce directories of such sites, a register of potential sites could be created to help spatial planners in selecting eventual future landfill sites.

Reviewer 2 Report

This is an interesting work showing how to delimit a new landfills using modern techniques to mitigate a set of geohazards and in compliance with public safety. The description of study subject, and the geology and geomorphology of the research area are  detailed and correct. However, there are some issues that should be solved by authors. The editorial side of the manuscript requires the biggest corrections. I have included all my comments in the comments to the attached PDF.

Author Response

POINT-BY-POINT RESPONSE TO REVIEWS

We are very grateful to the thorough revision by Reviewers, which helped us to clarify and, generally, to improve our early manuscript. We have considered carefully all the remarks and certainly appreciate the significant efforts to make our results more understandable to the readers. All the changes are highlighted in yellow in the revised manuscript to facilitate reviewer reassessment. Please, find below our responses to the reviews comments that are shown in red below each comment.

Response to Reviewer 2 Comments:

General comments: This is an interesting work showing how to delimit a new landfills using modern techniques to mitigate a set of geohazards and in compliance with public safety. The description of study subject, and the geology and geomorphology of the research area are  detailed and correct. However, there are some issues that should be solved by authors. The editorial side of the manuscript requires the biggest corrections. I have included all my comments in the comments to the attached PDF.

Response: Thank you very much for the careful and professional revision of our manuscript. We appreciate very much your precise work on improving our editorial side of the manuscript. We provide our point-by-point responses as follows:

Point 1: Line 47: I understand that this set of factors is quoted from existing literature. But it would be good to know – is it related to the law in Serbia, or was it selected by the authors of the manuscript?

Response 1: These factors are also included in the Law on Waste Management in the Republic of Serbia and are in accordance with criteria specified by Relevant criteria for determination the location of landfill site published in the Official Gazette of the Republic of Serbia No. 54/1992 and Regulation on waste disposal published in the Official Gazette of the Republic of Serbia No. 92/2010.

We added the above sentence at the end of the paragraph.

Point 2: Line 77: Add a space

Response 2: Corrected. Some spaces were probably disturbed during the conversion of word file to PDF in submission process.

Point 3: Line 99: You mentioned that some previous works exists aimed at landfill site selection in this region. Please shortly describe in this paragraph why do you think your studies are better/more complex/more advanced/using more complex point of view – than the previous studies? Authors described somewhat this problem in the whole Introduction, but this is “clue” of this work and should be emphasised here.

Response 3: We added more information in this paragraph on the aim of the study as follows:

Geological approach for landfill site selection used in this study provides more complex point of view than previous studies in this area due to the fact that geological criteria primarily control the suitability of landfill sites. Importance of bedrock is stressed as contaminated leachate may readily percolate downwards into the groundwater depending on geological setting. A new multidisciplinary approach is therefore needed for landfill site selection as geological criteria primarily control the suitability of candidate sites. It is important to highlight the fact that geological surveys need to be supported by the ecological and social considerations and reasonable site management. Thus, the best results in landfill planning may be achieved by the collaboration of spatial planners and geologists in order to ensure that all factors affecting the suitability of sites for landfill site are taken into consideration.

Point 4: Line 112: m a.s.l.?

Response 4: Corrected.

Point 5: Line 117: add a space

Response 5: Corrected.

Point 6: Lines 118-120: I understood this map only after help from google maps. It would be good to mark this area with an arrow (or not) and be sure to sign that it is Vršac.

Response 6: The position of Vršac municipality is marked in dark grey on a map of Serbia. In order to be more visible, we added a white arrow showing the position of Vršac municipality. Also, in caption to Figure 1 we added: (white arrow indicates the position of Vršac municipality).

Point 7: Line 124: add a space

Response 7: Corrected.

Point 8: Line 126: add a space

Response 8: Corrected.

Point 9: Line 124: add a space

Response 9: Corrected.

Point 10: Line 129: Put a comma

Response 10: Corrected.

Point 11: Line 131: add a space

Response 11: Corrected.

Point 12: Line 141: delete a space

Response 12: Corrected.

Point 13: Line 158: add a space

Response 13: Corrected.

Point 14: Line 185: add a space

Response 14: Corrected.

Point 15: Line 210: This empty space should be filled with text from below the Figure 3.

Response 15: Corrected. In original Word file prior to PDF conversion it was filled.

Point 16: Line2 203-206: Check spaces in this description. Should be unified.

Response 16: Corrected.

Point 17: The authors write ground-water and groundwater once. In the literal sence, it would be correct to write “ground water” everywhere.

Response 17: Corrected as ground water throughout the text.

Point 18: Line 219: add a space

Response 18: Corrected.

Point 19: Line 230: add a space

Response 19: Corrected.

Point 20: Line 274: add a space

Response 20: Corrected.

Point 21: Line 274: Dots are used everywhere in the text as decimal separators. It should be the same in the table.

Response 21: Corrected.

Point 22: Line 293

Response 22: Corrected.

Point 23: Line 298: Double space?

Response 23: Corrected.

Point 24: Line 322: With a capital letter?

Response 24: Corrected.

Point 25: Line 324: add a space

Response 25: Corrected.

Point 26: Line 392: Please check the references carefully as I found at least a few missing or not needed spaces. This should be unified according to the journal style.

Response 26: Checked and corrected.

Reviewer 3 Report

This manuscript reads well; I would like to proofread the original word document, if possible, but otherwise, it is a clear well-written manuscript. The figures are also particularly good. 

Author Response

POINT-BY-POINT RESPONSE TO REVIEWS

We are very grateful to the thorough revision by Reviewers, which helped us to clarify and, generally, to improve our early manuscript. We have considered carefully all the remarks and certainly appreciate the significant efforts to make our results more understandable to the readers. All the changes are highlighted in yellow in the revised manuscript to facilitate reviewer reassessment. Please, find below our responses to the reviews comments that are shown in red below each comment.

Response to Reviewer 3 Comments:

General comments: This manuscript reads well; I would like to proofread the original word document, if possible, but otherwise, it is a clear well-written manuscript. The figures are also particularly good. 

Response: We thank the reviewer for thoughtful review of our work and kind words. We have thoroughly reviewed the manuscript and corrected any errors we came across.

Reviewer 4 Report

Minor spelling and spacing issues exist in the paper.

Author Response

POINT-BY-POINT RESPONSE TO REVIEWS

We are very grateful to the thorough revision by Reviewers, which helped us to clarify and, generally, to improve our early manuscript. We have considered carefully all the remarks and certainly appreciate the significant efforts to make our results more understandable to the readers. All the changes are highlighted in yellow in the revised manuscript to facilitate reviewer reassessment. Please, find below our responses to the reviews comments that are shown in red below each comment.

Response to Reviewer 4 Comments:

General comments: Minor spelling and spacing issues exist in the paper. 

Response: We thank the reviewer for thoughtful review of our work and kind words. We have thoroughly reviewed the manuscript and corrected any errors we came across.

Round 2

Reviewer 1 Report

The authors have addressed all my comments.